# Unveiling the Metal-Dependent Aggregation Properties of the C-terminal Region of Amyloidogenic Intrinsically Disordered Protein Isoforms DPF3b and DPF3a

**DOI:** 10.3390/ijms232315291

**Published:** 2022-12-04

**Authors:** Tanguy Leyder, Julien Mignon, Denis Mottet, Catherine Michaux

**Affiliations:** 1Laboratoire de Chimie Physique des Biomolécules, UCPTS, University of Namur, 61 rue de Bruxelles, 5000 Namur, Belgium; 2Namur Institute of Structured Matter (NISM), University of Namur, 5000 Namur, Belgium; 3Namur Research Institute for Life Sciences (NARILIS), University of Namur, 5000 Namur, Belgium; 4Gene Expression and Cancer Laboratory, GIGA-Molecular Biology of Diseases, University of Liège, B34, Avenue de l’Hôpital, 4000 Liège, Belgium

**Keywords:** double-PHD fingers 3 (DPF3), intrinsically disordered protein, neurodegenerative diseases, aggregation, amyloid fibrillation, metal cations, spectroscopy, deep-blue autofluorescence, electron microscopy

## Abstract

Double-PHD fingers 3 (DPF3) is a BAF-associated human epigenetic regulator, which is increasingly recognised as a major contributor to various pathological contexts, such as cardiac defects, cancer, and neurodegenerative diseases. Recently, we unveiled that its two isoforms (DPF3b and DPF3a) are amyloidogenic intrinsically disordered proteins. DPF3 isoforms differ from their C-terminal region (C-TERb and C-TERa), containing zinc fingers and disordered domains. Herein, we investigated the disorder aggregation properties of C-TER isoforms. In agreement with the predictions, spectroscopy highlighted a lack of a highly ordered structure, especially for C-TERa. Over a few days, both C-TERs were shown to spontaneously assemble into similar antiparallel and parallel β-sheet-rich fibrils. Altered metal homeostasis being a neurodegeneration hallmark, we also assessed the influence of divalent metal cations, namely Cu^2+^, Mg^2+^, Ni^2+^, and Zn^2+^, on the C-TER aggregation pathway. Circular dichroism revealed that metal binding does not impair the formation of β-sheets, though metal-specific tertiary structure modifications were observed. Through intrinsic and extrinsic fluorescence, we found that metal cations differently affect C-TERb and C-TERa. Cu^2+^ and Ni^2+^ have a strong inhibitory effect on the aggregation of both isoforms, whereas Mg^2+^ impedes C-TERb fibrillation and, on the contrary, enhances that of C-TERa. Upon Zn^2+^ binding, C-TERb aggregation is also hindered, and the amyloid autofluorescence of C-TERa is remarkably red-shifted. Using electron microscopy, we confirmed that the metal-induced spectral changes are related to the morphological diversity of the aggregates. While metal-treated C-TERb formed breakable and fragmented filaments, C-TERa fibrils retained their flexibility and packing properties in the presence of Mg^2+^ and Zn^2+^ cations.

## 1. Introduction

Double-plant homeodomain (PHD) fingers 3 (DPF3) is a zinc finger protein and human epigenetic regulator found within the multiprotein BRM/BRG1-associated factor (BAF) chromatin remodelling complex. DPF3, also named BAF45c or CERD4, is a highly conserved member of the D4 protein family, which is characterised by the following domains (Figure 1): (i) an N-terminal 2/3 domain, (ii) a Krüppel-like C_2_H_2_ zinc finger (ZnF), and (iii) two C-terminal PHD ZnFs (PHD-1 and 2) [1]. This PHD tandem, being the only domain whose structure has been determined up to now, is able to recognise modified acetylated or methylated lysine residues on histone tails, allowing gene transcription through the recruitment of the BAF complex to genomic regions [2,3].

DPF3 plays an essential role in heart muscle cell development, and it has been found to be upregulated in patients suffering from heart hypertrophy, such as the Tetralogy of Fallot, a congenital cardiac malformation, consisting of right ventricular myocardium hypertrophy [4,5]. DPF3 is also involved in male infertility [6,7], colon defects (Hirschsprung’s disease) [8], as well as in many cancer types. DPF3 downregulation is indeed associated with the proliferation and motility of breast cancer cells [9]. In chronic lymphocytic leukaemia, a blood and bone marrow cancer, DPF3 upregulation leads to the proliferation of malignant cells [10]. DPF3 also maintains the stemness of glioma-initiating cells, which contributes to drug resistance, tumour recurrence, and growth in glioblastoma [11]. In addition, it appears to be a prognostic marker in lung cancer and chronic obstructive pulmonary disease [12]. Most recently, DPF3 has been found to be particularly essential to the signalling pathways involved in renal cell carcinoma (RCC) oncogenesis by inhibiting apoptosis, as well as promoting cell growth and RCC metastasis [13,14,15].

In humans, DPF3 actually exists as two splicing variants or isoforms, known as DPF3b and DPF3a. While their amino acid sequences remain identical from the N-terminus up to the 292nd residue, they differ in their molecular mass, C-terminus composition, and sequence length. More precisely, DPF3b contains 378 amino acids and has a theoretical molecular mass of 43.08 kDa, whereas DPF3a is 357 residues long and has a theoretical molecular mass of 40.24 kDa. Unlike DPF3b, which possesses the typical D4 family double-PHD fingers, DPF3a has a single truncated PHD ZnF (PHD-1/2), followed by the C-terminal domain (Figure 1). Although very little is known about the function of the DPF3a C-terminal region, it has been found to interact with the N-terminal intrinsically disordered region (IDR) of the SNIP1 protein in clear-cell RCC [15].

As unravelled in our latest studies, DPF3 is an intrinsically disordered protein (IDP) [16,17]. The two isoforms have a high disorder content, lack a hydrophobic core, and display an expanded and/or collapsed conformational ensemble. DPF3a, however, has proven to be more disordered than DPF3b thanks to its additional IDR at its C-terminus. Similar to other IDPs, such as α-synuclein (α-syn) or the tau protein, we also demonstrated that both DPF3 isoforms are prone to spontaneously aggregate into amyloid fibrils. Although they share a similar aggregate morphology, assembling into striated and twisted ribbon fibrils, their pathway and kinetics differ. While DPF3b progressively arranges into antiparallel β-sheets, DPF3a first forms α-helix intermediates before rearranging into β-sheets. In contrast with DPF3b, DPF3a rapidly aggregates, likely due to its highly dynamic C-terminal domain. Given all these properties, DPF3 isoforms are considered as new amyloidogenic IDPs [16,17].

It is well known that IDPs are highly involved in neurodegenerative diseases associated with protein misfolding and aggregation [18]. The progression of neurodegeneration is particularly affected by altered metal homeostasis. Indeed, deregulation of Cu^2+^, Zn^2+^, or Mg^2+^ levels in the brain induces neurodegeneration by an inflammatory response, oxidative stress, and accumulation of insoluble deposits [19,20]. Exposure to some metals, such as Ni^2+^, can also increase the incidence of neurodegenerative disorders [21]. Metal cations binding to IDPs stabilise folded states, notably by decreasing the electrostatic repulsions between negative charges. As aggregation processes arise from partially folded conformations, metal binding can either induce aggregation-prone conformers that will accelerate protein assembly or stabilise non-amyloidogenic folds [22,23,24]. Metal-dependent aggregation also relies on physicochemical factors, such as pH, temperature, protein, and metal ion concentration [25]. Furthermore, metal coordination to proteins is defined by several parameters, including the dielectric environment, solvent accessibility, and cation properties. Only a few amino acids, such as, histidine, cysteine, aspartate, and glutamate, are most involved in the binding [23].

Amongst the most common neurogenerative disorders are Alzheimer’s disease (AD) and Parkinson’s disease (PD). AD is characterised by amyloid plaques, resulting from the deposition of amyloid β peptide (Aβ) fibrils. Interactions with metal cations, such as Cu^2+^ and Zn^2+^, found in a high concentration at synapses, have been shown to impact the aggregation mechanism of Aβ. Indeed, Cu^2+^ binding accelerates the Aβ aggregation process and tends to form amorphous oligomers instead of fibrils. Zn^2+^ binding also leads to rapid aggregation of various oligomers that are less stable than those formed without metal ions [22,26,27]. Similarly, Zn^2+^ binding to the tau protein, whose aggregation is also a hallmark of AD, significantly accelerates its fibrillation, unlike Cu^2+^, which does not directly affect tau aggregation. Nevertheless, under reducing conditions, Cu^2+^ ions bound to tau fibrils participate in the generation of reactive oxygen species, leading to oxidative stress, cell damage, and the production of small toxic aggregates, contributing to neuron death [22,28,29,30]. Moreover, Ni^2+^ has been shown to modulate the conformation of tau, by preventing its fibrillation and forming short fragmented aggregates in AD [31].

Concerning PD, its pathogenesis is strongly related to α-syn aggregation into toxic oligomers, fibrils, and inclusion bodies, known as Lewy bodies. As with AD, it is recognised that metals have an impact on PD due to the modification of the α-syn conformation and aggregation pathway. Since Cu^2+^ has a high affinity for α-syn, the protein is bound to these cations in vivo, which accelerates aggregation without altering fibril morphology and forms neurotoxic oligomers [32,33]. On the other hand, at low concentrations, Mg^2+^ cations play a neuroprotective role in PD by inhibiting α-syn self-assembly and counteracting the effects of other metals [34]. In addition, Zn^2+^ binding decreases the risk of PD by promoting the formation of α-syn fibrils rather than oligomers, found to be more neurotoxic [22,35]. Zn^2+^ ions are also able to hinder α-syn aggregation through enhancing the chaperone activity of human serum albumin, which protects aggregation prone regions of α-syn [36].

Metal cations have therefore predominant effects on the conformation and aggregation mechanism of neurodegeneration-associated IDPs. In that context, DPF3 is a zinc finger amyloidogenic IDP, which is seemingly implicated in both AD and PD. Indeed, this protein contributes to the development of AD [37,38,39], and high expression levels of DPF3 have recently been detected in neuron clusters that are likely involved in cellular damage in PD [40]. Assessing the sensitivity of DPF3 to metals will help to elucidate its function in neurodegenerative diseases. DPF3 isoforms being differentiated by their C-terminus, which contains ZnFs (C_2_H_2_ and PHD domains), the metal-mediated aggregation properties of these regions have been investigated using an array of prediction tools and biophysical methods. In the present study, we report the disorder-associated features of DPF3b and DPF3a C-termini (from the 200th to the last amino acid; Appendix A), as well as the influence of various neurodegeneration-relevant divalent metal cations (Cu^2+^, Mg^2+^, Ni^2+^, and Zn^2+^) on their aggregation propensity and pathway.

## 2. Results

### 2.1. DPF3 C-terminal Regions Have Intrinsic Disorder Properties

Because DPF3b and DPF3a were identified as two IDPs presenting different IDRs in their respective C-terminal (C-TER) end, the disorder content of these C-TER regions has been investigated by combining prediction tools and spectroscopic techniques. As expected from the truncated PHD-1/2 and the subsequent IDR of DPF3a, its C-terminal region (C-TERa) is considered mainly disordered with an overall predicted per-residue percentage of intrinsic disorder of 76%. On the other hand, the presence of the PHD tandem in the DPF3b C-terminus (C-TERb) significantly decreases its disorder content to 35%, corresponding to the small IDR between the C_2_H_2_ and PHD-1 domains. Although C-TERb disorder score seems low, it can still be seen as a partially disordered domain. This is in good agreement with the proportion between disorder- and order-promoting residues [41]. C-TERa is indeed more enriched in disorder-promoting residues (60%) than C-TERb (48%), as well as depleted in order-promoting residues (22% and 35% for C-TERa and C-TERb, respectively).

The disorder–order discrepancy between the two C-TER regions is also highlighted by the cumulative distribution function and the charge–hydropathy plots. On the cumulative distribution function plot, a protein is expected to be mostly ordered if its curve is above the boundary and disordered if it is below [42]. In that respect, C-TERb appears mainly ordered, while C-TERa is disordered (Appendix A). Regarding the charge–hydropathy plot, it allows discriminating the IDPs from ordered proteins based on their absolute mean net charge (<R>) and mean hydropathy (<H>). Proteins found above the boundary line, defined by <R> = 2.785<H> − 1.151, are considered to be mostly disordered, while those located below are expected to be ordered [43]. Whilst both C-TERs exhibit a low mean net charge, C-TERa has a lower hydropathy score, placing it amongst the IDPs (Appendix A). On the other hand, C-TERb is close to the boundary line, which is consistent with its more ordered nature.

In order to support these results, the conformational ensemble adopted by each C-TER domain was examined on Das–Pappu phase diagrams, which look at the distribution of the fraction of negatively charged residues against the fraction of positively charged ones [44,45]. C-TERb is found within the R1 area, while C-TERa is located in the R2 region (Appendix A). The R1 region is composed by weak polyelectrolyte IDPs (~25%) with globule- or tadpole-like conformations, harbouring a globular head and a disordered tail [46]. This is consistent with C-TERb being characterised by ordered domains separated by an IDR. The R2 region corresponds to IDPs (~40%) whose conformations are context-dependent, either collapsed or expanded. This is also consistent with C-TERa being more disordered and flexible than C-TERb. In addition, patterning parameters can be extracted from the Das–Pappu diagrams, namely κ and Ω. The κ parameter describes the distribution of charged residues within the sequence, while Ω includes proline (Pro) residues in the patterning. These values are scaled from 0 to 1, and the closer a protein is to 0, the more extended its structure is expected to be [44,47]. C-TER domains have comparable patterning parameters, indicating that expanded conformations may be favoured. κ values of 0.24 for C-TERb and 0.28 for C-TERa suggest that C-TERa structure is slightly more collapsed due to electrostatic interactions. On the contrary, the lower Ω value of C-TERa (0.17) shows that expansion-promoting Pro residues are more spread in the sequence in comparison with C-TERb (Ω = 0.23).

From these prediction data, C-TERb and C-TERa are assumed to display distinctive structural behaviours in vitro, which were investigated by absorption and emission spectroscopy. Similar to the DPF3 FL isoforms, both C-TER domains have their maximum absorption wavelength (λ_abs_) at 258 nm rather than 280 nm (Appendix A). Regarding C-TERb, containing three Trp residues in its PHD tandem, its low λ_abs_ can be explained by its relatively expanded conformation. Indeed, it has been shown that, upon protein unfolding, the contribution at 258 nm increases, reducing the foldedness index, determined from absorbance ratios at 280, 275, and 258 nm [48]. The IDPs are thus expected to yield a low foldedness index (<2.5) in their “native” state, which is the case for C-TERb and C-TERa with a foldedness index of 1.2 and 1.1, respectively.

To assess the presence of secondary structure elements in the C-TER regions, far-UV circular dichroism (CD) spectroscopy was used. The C-TERb spectrum displays a strong positive band at 202 nm, a slight negative band at 219 nm, and a more pronounced one at 225 nm, which are associated with β-sheets (Figure 2). A shoulder is also noticeable at 208 nm, which seems indicative of some α-helix structures. The negative band at 212 nm likely arises from both α-helix and β-sheet contributions. Although C-TERa has a distinct footprint, its CD spectrum reveals similar signatures related to antiparallel β-sheets with a positive band at 200 nm and a negative one at 225 nm (Figure 2). The latter may also emanate from disorder as unordered polypeptides have been shown to exhibit a broad negative band in the 220–230 nm range [49]. Nonetheless, a mix of α-helix and random coil is presumably responsible for the broad minimum observed between 206 and 210 nm [50]. Moreover, the overall low intensity of C-TERa spectrum clearly suggests a lack of a regular structure or ordered elements [51]. BeStSel estimations are in good agreement with the higher proportion in α-helices and β-sheets for C-TERb (Table 1), as expected from its C_2_H_2_ and PHD ZnFs. Consistent with the discrepancy between the CD spectra, C-TERa has more random coil.

The local environment and exposition state of aromatic residues were probed by intrinsic tryptophan (ITF) and tyrosine (ITyrF) fluorescence to further examine the (un)folding state of each C-TER. Trp residues are solvatochromic fluorophores, emitting from 308 to 355 nm when found in a non-polar (low emission wavelength, λ_em_) or very polar (high λ_em_) environment [52,53]. Although Tyr residues are less sensitive to polarity, their excitation leads to a characteristic emission band in the 300–310 nm range when there is no fluorescence resonance energy transfer (FRET) to Trp residues [54,55]. Looking at Tyr fluorescence has proven to be very structurally informative, especially for IDPs and Trp-depleted proteins, which is the case for C-TERa (Figure 1).

C-TERb contains three Trp residues: one in PHD-1 and two in PHD-2 (Figure 1). Regarding its ITF spectrum, the emission band is centred at ~334 nm, which is typical of Trp residues partially exposed to the solvent and/or polar residues (Figure 3A). C-TERb Trp residues in PHD ZnFs are therefore not fully buried, but rather edge- and/or one-face-exposed. The ITyrF spectra of C-TERb also exhibit an emission band at ~334 nm with a very slight shoulder at 305–310 nm (Figure 3B). With respect to ITF, the absence of a shift in the ITyrF λ_em_ suggests that Trp-Tyr FRET occurs, leading to the observation of Trp fluorescence. This is consistent with the distribution of Tyr residues in C-TERb sequence. Indeed, three out of its seven Tyr residues are found in the vicinity of three Trp residues in PHD-1 and -2. Although Trp emission is the majority, the slight shoulder at lower wavelengths relates to Tyr fluorescence, presumably coming from the three Tyr residues in the C_2_H_2_ domain [54,56].

In comparison, C-TERa ITyrF λ_em_ is located at ~328 nm with a more pronounced shoulder towards 305 nm, as evidenced by the difference spectrum (Figure 3B). Because C-TERa does not contain any Trp residues, the overall red shift likely arises from tyrosinate formation due to proton transfer between the tyrosyl side chain hydroxyl moiety and an aspartyl (Asp) or gultamyl (Glu) carboxylate acceptor within the primary structure [57]. Similar to C-TERb, the shoulder at 305 nm can also be attributed to the Tyr residues in the C_2_H_2_ ZnF, whose fluorescence becomes more visible due to the absence of Trp residues.

### 2.2. DPF3 C-terminal Regions Are Prone to Aggregate In Vitro

Recently, we demonstrated that both full-length (FL) DPF3 isoforms are able to aggregate in vitro with different kinetics into morphologically similar amyloid fibrils [17]. Herein, the aggregation propensity of C-TERb and C-TERa was also spectroscopically investigated over an incubation period of four days (96 h) at ~20 °C. Protein samples were kept in the same buffer (TBS) and at a concentration of ~10 µM. As amyloidogenic proteins are known to be enriched in β-sheets during their aggregation process, secondary structure modifications were monitored by far-UV CD spectroscopy (Figure 4, Table 1).

Over the course of only a few days, C-TERb undergoes dramatic conformational rearrangements (Figure 4A). After 24 h, the α-helix component increases with the apparition of a slight negative band at 222 nm and an overall intensity increase in the 206–210 nm region. From spectrum deconvolution, this is accompanied by a loss in antiparallel β-sheets and turns. In the next 24 h, the CD signature is completely shifted towards a single broad minimum centred at ~227 nm, and a very subtle shoulder at 212 nm remains. Over the next two days, this minimum sharpens and significantly increases in intensity. This minimum relates to a typical enrichment into β-sheet structures. Curiously, at 72 h of incubation, there is the emergence of a positive shoulder at 207 nm, which disappears after 96 h. Interestingly, secondary structure content estimations reveal that C-TERb progressively transitions to parallel β-sheets and turns at the expense of antiparallel β-sheets. The coexistence of antiparallel and parallel β-sheets may explain the minimum broadening. The formation of parallel β-sheets is also a marker of amyloid aggregation, and a mix with antiparallel β-sheets has also been reported in fibrils [58,59,60,61].

C-TERa exhibits comparable conformational transformations (Figure 4B), except for the first 24 h. Indeed, instead of a gain, the α-helix and random coil contents decrease, as visualised by the loss of the minimum between 206 and 210 nm. An enrichment in antiparallel β-sheets is also observed. No significant structural modification seems to occur after 48 h. This is supported by the BeStSel estimations, showing that only a small fraction of turns is changed into coil. In the next 48 h of incubation, C-TERa signature is very similar to that of C-TERb with the emergence of the minimum at 227 nm and the positive increase at 202 nm. Spectrum deconvolution describes the same tendency, consisting of a progressive enrichment into parallel β-sheets and turns, as well as a decrease in antiparallel β-sheets. The positive shoulder at 207 nm, detected for C-TERb after 72 h, is not observed on C-TERa spectrum.

Near-UV CD spectroscopy was used to look at tertiary structure transformations during C-TER transition into β-sheets (Appendix A). The near-UV spectrum can be divided into three main absorption regions attributed to aromatic residues: (i) phenylalanine (Phe) between 250 and 270 nm, (ii) Tyr between 270 and 280 nm, and (iii) Trp between 280 and 300 nm [62]. The great variability amongst the obtained curves suggests that the tertiary structure and the aromatic residue environment of each C-TER are modified to a large extent. Regarding C-TERb, little difference is observed after 24 h with a small overall increase in intensity (Appendix A). At 48 h, the near-UV footprint considerably changes with a large intensity increase in the 270–300 nm range. The successive intensity decrease and increase, as well as shifts of this band at 72 and 96 h, respectively, show that structural transformations are ongoing. Strikingly, though there are no Trp residues in C-TERa, the latter appears to follow the same spectral pattern from 48 to 96 h of incubation, that is the presence of the broad positive band between 270 and 300 nm (Appendix A). In that respect, such a signal likely arises from Tyr rearrangement upon amyloid core formation. Complementarily, the appearance of the positive band between 48 and 96 h is indicative of the ordering of hydrophobic residues [63,64]. Interestingly, this near-UV CD band could also be an amyloid fingerprint. Indeed, a similar behaviour has already been detected for a non-amyloid-β component (NAC) (1–13) [65]. Over the course of fibrillation, NAC (1–13) displays a gradual intensity increase at 285 nm, even though it does not contain any aromatic residues.

The modification of C-TERb and C-TERa tertiary structure upon aggregation was further and locally examined by ITF and ITyrF. With respect to ITF, the Trp residues of C-TERb exhibit an increase in fluorescence intensity at ~334 nm after 96 h (Figure 5A), without showing any shift relating to significant conformational modifications (Figure 5B). This absence of the λ_em_ shift combined with the intensity increase indicates that some Trp residues are still partially exposed to polar amino acids or aqueous solvent, and some are in a more hydrophobic environment. No extensive modification in the Trp environment is consistent with the near-UV data (Appendix A). Therefore, the Trp residues, and consequently PHD-1 and 2 ZnFs, are seemingly not involved in the formation of the amyloid core. Nevertheless, just after 24 h, a remarkable second emission band appears at ~456 nm, whose intensity gradually increases over time.

These dual emission modes have been associated with amyloidogenic proteins and, most notably, have been reported for transthyretin and DPF3 isoforms [16,17,66,67]. Although the mechanism underlying this phenomenon, referred to as deep-blue autofluorescence (dbAF), remains debated, many studies propose that it arises from low-energy transitions caused by electron delocalisation through hydrogen bonds within the highly stable β-sheet network in amyloid aggregates [68,69,70]. Other dbAF contributors have recently been identified, namely supramolecular packing, π-π stacking of aromatic residues, or Tyr oxidation, which makes it more difficult to rationalise [71,72,73,74]. Due to the large diversity in sequence composition, the formation of dbAF-emitting fluorophores is most likely protein-specific, resulting in a vast array of unique λ_em_ in the visible range. Still, dbAF is a sensitive and reliable intrinsic fluorescence method for probing structural changes, especially for aggregation-prone proteins [17,67]. 

The dbAF band is also visible, slightly shifted to ~460 nm, on the ITyrF spectrum of C-TERb (Figure 5C), suggesting that, along with Trp, the Tyr residues participate in dbAF. In addition, while the first emission band remains at 334 nm, the small shoulder at ~305 nm completely disappears after 96 h, as highlighted by the difference spectrum (Figure 5D). This loss likely relates to an increase in Trp-Tyr FRET due to conformational rearrangement. As observed on the C-TERb ITF spectrum (Figure 5A), there is an overall intensity increase of the first emission band over time, which can be indicative of Tyr residues found in a more hydrophobic environment. dbAF emission at 460 nm emerges on the C-TERa ITyrF spectrum in the first 24 h as well and considerably increases in intensity until it becomes the majority band (Figure 5E). Regarding the C-TERa Tyr emission band, the shoulder between 300 and 310 nm is gradually lost and λ_em_ is red-shifted to ~333 nm after 96 h of incubation (Figure 5F). In the absence of Trp residues, these two phenomena are associated with aggregation-driven structural rearrangements, leading to Tyr residues close to Asp or Glu proton acceptors. Similar to C-TERb, the band intensity increase also suggests that the Tyr residues are organised in a more hydrophobic core.

In order to better decipher dbAF contributions, the excitation spectra (λ_em_ = 456 nm) of C-TERb and C-TERa were recorded (Appendix A). The two C-TER regions have similar spectral profiles consisting of three distinct excitation bands, whose intensities progressively increase over the course of aggregation. The first band at 237 nm corresponds to the absorption of peptide bonds [75,76]. The second one, which is broader and centred at ~270 nm, is attributed to aromatic residues, explaining the appearance of the dbAF emission band on the ITF and ITyrF spectra. The third band at 400–405 nm is the main dbAF component. Interestingly, C-TERb is characterised by a very slight fourth excitation band between 325 and 330 nm, which could originate from unique infrastructure interactions within its amyloid core (Appendix A). 

By exciting each C-TER at 400 nm, intense dbAF-specific emission signals at 456–460 nm were obtained (Appendix A). Plotting dbAF intensity at 456 nm against the incubation time gives rise to a typical sigmoid-like kinetic curve for the two C-TER isoforms (Figure 6A). Such a three-phase kinetic profile is well described in the literature and is characteristic of amyloidogenic proteins [77,78]. The lag phase, corresponding to structural transitions, nucleation, and assembly of oligomers, is nearly the same for the two proteins: 36 and 40 h for C-TERb and C-TERa, respectively. The growth or elongation phase, that is oligomer clustering and protofibril formation, starts with a rapid increase in fluorescence intensity. A remarkable difference is that the stationary phase (mature fibrils) is reached after 72 h for C-TERb, while C-TERa hits it after 90 h. Indeed, C-TERb is characterised by a shorter growth phase of ~36 h, suggesting that the aggregation process, more precisely the elongation step, takes less time for this isoform. Nevertheless, the intensity associated with the plateau is approximately three-times higher for C-TERa than C-TERb, which may explain its longer growth phase (of ~50 h).

In order to further ascertain the amyloid nature of C-TER aggregates, as well as validate dbAF robustness in distinguishing fibrillation phases, incubated C-TER isoforms were tested for thioflavin T (ThT) binding. The ThT binding assay is commonly used to kinetically monitor amyloid fibrillation. Indeed, this extrinsic fluorophore selectively interacts with amyloid fibrils, resulting in an emission increase at 485 nm after excitation at 440 nm [79]. For each C-TER, this typical band increase is observed over time (Appendix A). However, the intensity remains quite low in the first 24 h, meaning that non-amyloid aggregates are first mainly formed. The formation of such species corresponds to the lag phase. Similar to dbAF, sigmoid-like kinetic curves were obtained by plotting the ThT fluorescence intensity with respect to the time (Figure 6B). dbAF and ThT tendencies are reasonably comparable, which include a longer and more pronounced growth phase for C-TERa, as well as a plateau of higher intensity in the stationary phase. Furthermore, this phase is also completely reached after 72 and 96 h for C-TERb and C-TERa, respectively. 

### 2.3. Transmission Electron Microscopy Reveals the Morphologic Diversity of DPF3 C-terminal Region Aggregates 

In order to visualise the morphology of C-TERb and C-TERa aggregates, samples were negatively stained with uranyl acetate and examined by transmission electron microscopy (TEM). TEM micrographs were recorded after 144 h (6 days) of incubation at ~20 °C in TBS for each C-TER (Figure 7). In both cases, fibrillar structures were expected to be detected, as the ThT binding was positive after a few days (Appendix A). Indeed, the two C-TER isoforms assemble into long amyloid filaments upon aggregation. Remarkably, there is no morphologic discrepancy between C-TER isoforms, both characterised by a limited diversity in aggregate forms and shapes. The major population of aggregates consists of single straight fibrils (SSFs), with a width varying from 15 to 20 nm (Figure 7A,E), which is in the typical range observed for mature amyloid fibrils [80]. When found close to each other, SSFs are also able to pack in pairs, resulting in double straight fibrils (DSFs) that do cross each other (Figure 7A,E). Occasionally, SSFs and/or DSFs cluster into isolated wider structures, called multiple straight fibrils (MSFs) (Figure 7B,F). In MSFs, each individual SSF is clearly visible. The formation of DSFs and MSFs could be associated with the maximisation of fibril stability. Instead of packing, SSFs are more commonly found in extended and open fibrillar networks in which SSFs intertwine, suggesting the occurrence of secondary nucleation points on the grown fibrils (Figure 7C,G). Much less frequently, thin twisted fibrils (TTFs) have been spotted (Appendix A). Beyond ordered fibrous aggregates, large amorphous phases were also observed in incubated samples (Figure 7D,H).

### 2.4. Metal Cations Have Distinct Effects on the Aggregation of DPF3 C-terminal Regions 

Because metals are potent conformational modulators of amyloidogenic proteins, the influence of various divalent metal cations (M^2+^), that is Cu^2+^, Mg^2+^, Ni^2+^, and Zn^2+^, on C-TERb and C-TERa aggregation was studied by combining spectroscopy and microscopic techniques. Each C-TER, at a concentration of ~5 µM, was incubated during 96 h in TBS at ~20 °C in the presence of 100 µM of the selected metal ions (1:20 protein:metal ratio). The presented data are focused on the third day (72 h) of incubation. Modifications in β-sheet enrichment were assessed by far-UV CD spectroscopy (Figure 8, Table 2).

At first glance, metal binding does not particularly impact C-TERb conformational rearrangement (Figure 8A). Indeed, the CD spectra have comparable shapes and band intensities, especially for Mg^2+^ and Zn^2+^, meaning that the formation of β-sheets is not impaired by the presence of metal ions. This is supported by spectral deconvolution, revealing no major conversion in turn and parallel β-sheet contents (Table 2). Nevertheless, all metals appear to promote the formation of antiparallel β-sheets, with Ni^2+^ having the strongest effect. In addition, in the presence of Cu^2+^, the minimum is shifted from 227 to 224 nm and the positive band from 202 to 200 nm. While this cannot be attributed to the emergence of new secondary structure elements, this spectral change could arise from specific metal-bound structural motifs.

More variations were observed for C-TERa (Figure 8B). While the absorption minimum remained at 227 nm for Mg^2+^ and Ni^2+^, it is shifted to 225 nm with Cu^2+^ and further displaced to 221 nm in the presence of Zn^2+^. Once again, peculiar metal-induced folds could be responsible for such spectral shifts. The positive band absorption is also slightly shifted to 200 nm for Cu^2+^ and 201 nm for Mg^2+^ and Ni^2+^. There are no notable variations in the negative band intensity, except for Mg^2+^. Indeed, Mg^2+^ noticeably increases the negative band intensity of C-TERa at 227 nm. In the absence of metals, such an increase was associated with the formation of fibrils and to an enrichment into parallel β-sheets (Figure 4B), which is consistent with the estimated secondary structure content (Table 2). In that respect, Mg^2+^ seems to act as a C-TERa aggregation enhancer. Similar to C-TERb, Ni^2+^ and Zn^2+^ increase the proportion of C-TERa antiparallel β-sheets. Cu^2+^ does not significantly impact C-TERa secondary structure.

Given that the secondary structure content of aggregating C-TER remains globally unaffected by metals, the protein conformational state was first verified by near-UV CD spectroscopy (Appendix A). Regarding C-TERb, the tendencies observed in near-UV are in good agreement with far-UV CD data (Appendix A). Without metal, as well as in the presence of Mg^2+^, Ni^2+^, and Zn^2+^, C-TERb spectra display a broad positive band in the Trp region between 290 and 310 nm, with some intensity discrepancy. As a reminder, this band may have an amyloid origin, as it was already detected in Trp-free fibrillated proteins. On the other hand, this band is less pronounced with Cu^2+^, and a noticeable positive band appears at around 270 nm with a shoulder at ~260 nm. These two features show that Cu^2+^ binding to C-TERb alters the organisation of Tyr residues more, while keeping the β-sheets, which is consistent with the spectral shifts in far-UV CD (Figure 8A). For every metal, numerous changes are also visible in the region between 250 and 270 nm, suggesting different structural transformations at the level of Phe residues upon metal addition.

With respect to C-TERa, near-UV data are also coherent with far-UV CD spectra (Appendix A). As for C-TERb and metal-free C-TERa, samples incubated with Mg^2+^ and Ni^2+^ exhibit a similar broad positive band between 290 and 310 nm. Its intensity is noticeably increased in the presence of Mg^2+^, which could correlate with the stronger β-sheet signal obtained in far-UV (Figure 8B). Therefore, neither Mg^2+^ nor Ni^2+^ seem to disrupt the amyloid core. Cu^2+^ and Zn^2+^ have a deeper impact on C-TERa tertiary structure. On the Cu^2+^ spectrum, the positive band is shifted to 270–280 nm, while a second band appears for Zn^2+^ at ~275 nm. Cu^2+^ and Zn^2+^ cations are therefore hypothesised to modulate the environment of Tyr residues. Moreover, compared to C-TERb, less variations are observed in the Phe region. 

M^2+^ cations influence on C-TERb and C-TERa tertiary structure upon aggregation was also reviewed by ITF and ITyrF. Regarding C-TERb spectra, metal addition does not affect the position of the Trp emission band with the λ_em_ remaining at ~334 nm (Figure 9B). This is still indicative of Trp residues partially exposed to a polar environment. The Trp emission band intensity is nonetheless decreased with each metal cation, suggesting that metal binding induces the rapprochement of quenching groups in the close neighbourhood of Trp residues. However, metals strongly impact the dbAF band (λ_em_ = 456–460 nm) by reducing its intensity in the following order: Ni^2+^ > Cu^2+^ > Zn^2+^ > Mg^2+^ (Figure 9A). Indeed, the dbAF signal completely disappears in the presence of Ni^2+^ cations and is almost halved with Mg^2+^ compared to metal-free C-TERb. A similar behaviour is observed on ITyrF spectra with a second band still at ~456 nm and whose intensity decreases in the same manner as for ITF (Figure 9C). Moreover, there is still no shift of the first band (~334 nm) and the shoulder at ~305 nm, initially present at 0 h (Figure 5D), disappeared with each metal (Figure 9D). Interestingly, Cu^2+^ is the only metal that induces a significant intensity decrease of the first emission band, suggesting either that Tyr residues are found in a less hydrophobic environment or that Cu^2+^ ions interact with Tyr residues.

On the other hand, the behaviour is completely different for the C-TERa ITyrF spectra (Figure 9E). Indeed, while the first emission band remains at the same wavelength (~333 nm) and does not present any shoulder whatever the condition (Figure 9F), the second band is affected by metal binding. Apart from Zn^2+^, only its intensity is modified. While the dbAF signal is considerably extinguished by the addition of Cu^2+^ and Ni^2+^, Mg^2+^ cations increase its intensity by a factor of two. Interestingly, incubation with Zn^2+^ induces not only an overall intensity increase, but also a dramatic shift of the dbAF band from ~460 to ~490 nm. Previous reports have suggested that external factors, such as metal ions, could modify the dbAF properties of aggregating proteins [81]. In addition, the C-TERa Tyr emission band is notably decreased in intensity with Cu^2+^ and Zn^2+^, implying either metal-induced fluorescence extinction or quenching moieties getting closer to Tyr residues upon metal binding. As for C-TERb, Tyr residues may also be found in a less hydrophobic environment.

Excitation spectra were also recorded at a λ_em_ of 456 nm to assess any changes in dbAF contributions (Appendix A). For C-TERb, no spectral modifications due to metal binding are observed. Four excitation contributions are still identified: a slight shoulder between 325 and 330 nm and three bands at 237, 270, and 400 nm (Appendix A). These bands are barely visible with Ni^2+^ and less distinguishable with Cu^2+^ and Zn^2+^. Apart from Zn^2+^ and Cu^2+^ binding, the results are similar for C-TERa (Appendix A). Indeed, while the addition of Cu^2+^ induces a new excitation band at ~300 nm (Appendix A), Zn^2+^ causes the shift of the main dbAF contribution from 400–405 to 410 nm (Appendix A). As Zn^2+^-bound C-TERa has its dbAF emission band shifted to 490 nm (Figure 9E), an excitation spectrum was also recorded at this λ_em_ (Appendix A). While the first two excitation bands are still located at the same wavelengths, the major band appears at ~415 nm, and interestingly, a slight shoulder at ~340 nm emerges. This suggests that the C-TERa dbAF properties are strongly affected by the addition of Zn^2+^, which is not the case with the other metals tested.

Considering dbAF has proven to be as efficient as ThT binding assays to describe the aggregation kinetics, the evolution of autofluorescence at 456 nm in the presence of M^2+^ cations was monitored over time, namely for 96 h (Figure 10). First, the two C-TER isoforms unexpectedly displayed shorter lag (of ~24 and ~30 h for C-TERb and C-TERa, respectively) and elongation (of ~24 h) phases at lower concentrations, from 10 (Figure 6) to 5 µM (Figure 10). Aggregation kinetics are known to be concentration-dependent. Nucleation events (lag phase) are usually shorter in the presence of a higher protein concentration due to the occurrence of more protein–protein interactions [82], but the opposite trend is curiously observed here. As expected from the ITF (Figure 9A), ITyrF (Figure 9C), and dbAF (Appendix A) spectra, the selected metals have an inhibitory effect on C-TERb fibrillation (Figure 10A). Amongst them, Ni^2+^ has the strongest one by completely shutting down C-TERb autofluorescence. This phenomenon is directly related to aggregation and metal-induced structural changes, not only because Ni^2+^ cations do not extinguish the dbAF signal and ThT emission on preformed C-TER fibrils, but also because they do not interfere with ThT fluorescence in metal-only samples. These observations are also true for all the metals tested. The overall low dbAF signal of Ni^2+^-exposed C-TERb suggests that barely any fibrils are formed, which is supported by the absence of a significant increase in ThT emission (Appendix A). Therefore, Ni^2+^ is hypothesised to favour and stabilise less autofluorescent and ThT-negative oligomers, resulting in a weak and extended growth phase of ~66 h. The lag phase is slightly increased to ~30 h compared to C-TERb without metals. Cu^2+^ and Zn^2+^ have comparable kinetic plots, presenting a longer lag time of ~36 h, but a growth phase of ~26 h, similar to that of metal-free C-TERb. The very low intensity of their plateau also argues for a lower fibril/oligomer or fibril/monomer ratio, which is consistent with their relative ThT fluorescence (Appendix A). The addition of Mg^2+^ only reduces the stationary phase level, suggesting, along with ThT data, that fewer fibrils are formed.

Regarding C-TERa, metals have more varied effects on its dbAF emission (Appendix A) and kinetics (Figure 10B). Both Ni^2+^ and Cu^2+^ have a consequent impact on its fibrillation pathway by extending the lag and elongation times to ~50 h and >96 h, respectively. As for C-TERb, this is accompanied by a low intensity fluorescence signal in the stationary phase. The stabilisation and assembly of weakly fluorescent oligomers or aggregates could thus be promoted by Cu^2+^ and Ni^2+^. This explanation is reinforced by their respective low ThT signal (Appendix A). Contrary to C-TERb, Mg^2+^ cations have an enhancement effect on C-TERa aggregation, which substantiates the far- and near-UV CD results (Figure 7B and Appendix A). While the lag and growth phases remain mainly unaffected, the plateau intensity is significantly increased, indicating a higher fibril proportion and/or the formation of structurally more emitting species; an intense ThT emission band is coherently observed (Appendix A). When incubated with Zn^2+^, the C-TERa lag phase is extended to approximately 44 h, its stationary phase only being reached after 60 h. It can be noted that Zn^2+^ cations present the shortest growth phase. The longer lag phase likely corresponds to the generation of structurally different Zn^2+^-mediated fibrillar aggregates as the ThT assay ascertains the formation of fibrils (Appendix A). In addition, dbAF monitoring at the maximum λ_em_ (490 nm) of Zn^2+^-bound C-TERa results in the same kinetic trace.

Given the effects of metal cations on the aggregation kinetics and dbAF signatures, TEM was used to investigate the emergence of unique morphological features that may relate to the spectral changes described (Figure 11, Appendix A). As without metals, TEM micrographs were recorded after 144 h (6 days) of incubation at ~20 °C in TBS. In the presence of Cu^2+^, C-TERb forms long and bent single fibrils of 16 nm in width (Figure 11A and Appendix A). Although some flexibility seems to be retained, several breaking points appear along the length of the fibrils. Such a weakness raises the risk of the rupture of the fibrillar structure, which is indeed observed for C-TERa incubated with Cu^2+^ (Figure 11E and Appendix A). C-TERa fibrils bound to Cu^2+^ are larger than those of its isoform (20 to 28 nm wide) and are found unevenly fragmented into shorter ones. While the C-TERb-Cu^2+^ filaments have also a more striated or fibrous aspect, the C-TERa-Cu^2+^ SSFs look slightly twisted. Interestingly, those C-TERa fragments do not act as amyloid seeds, as no further increase in ThT or dbAF emission nor second elongation phase was later observed (at 144 h of incubation). The same applies to C-TERb. Partial disaggregation and interruption of the fibrillar structure could lead to the accumulation of less dbAF-emitting and ThT-positive species, resulting in the reduction of amyloid-related fluorescence (Figure 10).

This hypothesis is also supported by the micrographs obtained for C-TERb treated with Mg^2+^ (Figure 11B and Appendix A), Ni^2+^ (Figure 11C and Appendix A), and Zn^2+^ (Figure 11D and Appendix A). For each of these metals, C-TERb aggregates into 18–20 nm-wide SSFs broken in several places. Albeit that the ability to cluster into MSFs is somehow conserved for C-TERb-Ni^2+^ fibrils, they present critical breaking points, conducting MSFs to disassemble. Thinner fibrils (10 nm wide) are less frequently observed. The emergence of breaking points is a corollary of the metal-induced embrittlement and rigidification of fibrils. This is particularly noticeable for C-TERa-Ni^2+^ fibrils, manifesting themselves as shorter, brittle, and needle-shaped aggregates (Figure 11G and Appendix A). Wider fibrous fragments (~40–45 nm wide) are also detected, probably generated by the fragmentation of DSFs or MSFs.

Congruent with C-TERa spectroscopic behaviours, distinct aggregation patterns are disclosed in the presence of Mg^2+^ and Zn^2+^ cations. Mg^2+^-mediated fibrillation produces long and undulating 18 nm-wide SSFs, which look much like metal-free C-TERa fibrils (Figure 7) and whose flexibility, as well as tendency to stack into DSFs are preserved (Figure 11F and Appendix A). Analogous morphological features are observed for Zn^2+^-induced aggregation (Figure 11H and Appendix A). Very much like C-TERa in the absence of metals (Figure 7F), the C-TERa-Zn^2+^ SSFs tend to clump into densely packed MSFs. Nevertheless, there is a noteworthy difference in terms of fibril dimensions. In every other condition, the width of individual fibrils is in a typical 15–20 nm range, whereas the C-TERa-Zn^2+^ SSFs are only 5 to 8 nm wide. The unique dbAF fingerprint of C-TERa incubated with Zn^2+^ could have a structural origin, which is arguably reflected by this width divergence. In addition, the two C-TER isoforms also present amorphous aggregates for each metal condition (Appendix A).

## 3. Discussion

DPF3 is an amyloidogenic IDP that has been identified in many cancer types, as well as in neurogenerative disorders, such as AD and PD. The two different isoforms, DPF3b and DPF3a, have distinct C-TER regions. While C-TERb contains three ZnFs (C_2_H_2_, PHD-1 and -2), C-TERa has the C_2_H_2_ and a truncated PHD tandem (PHD-1/2), as well as a disordered C-terminal domain. In the present study, we reported the structural characteristics and the aggregation propensity of each C-TER. Expectedly, the two C-TER isoforms are structurally different. From sequence-based prediction data, C-TERa is more disordered, enriched in disorder-promoting residues, and adopts a more expanded conformational ensemble than C-TERb. Nevertheless, C-TERb also exhibits disorder properties. Such discrepancies arise from the disordered C-terminal domain of C-TERa and are also reflected in the spectral signatures.

Although C-TERb contains three Trp residues in its double-PHD domain, while C-TERa none, they both have their λ_abs_ located at 258 nm, suggesting that aromatic residues are not found in a highly hydrophobic environment. This is not surprising for C-TERa, as Trp-depleted proteins, such as α-syn, have been shown to have their λ_abs_ shifted to 258 nm [83]. Regarding C-TERb, ITF revealed that its Trp residues are partially exposed to the aqueous solvent or polar amino acids. Most C-TERb Tyr residues have been shown to be in the vicinity of Trp residues, allowing Trp-Tyr FRET. In comparison, C-TERa Tyr emission is partially red-shifted, which is likely due to the tyrosinate formation of Tyr-261. Given its proximity with Asp-263, Tyr-261 could indeed be involved in hydrogen bonding (Figure 1 and Figure 12). Albeit that this is usually observed at a very basic pH, such tyrosinate emission has already been reported for other Trp-depleted protein in physiological conditions. It was assigned to Tyr residues placed in a tyrosinate-facilitating environment by the protein tertiary structure [54,84]. Tyr-261 tyrosinate emission could also be present for C-TERb, but overlapped by the Trp-Tyr FRET signal.

Regarding the secondary structure content, C-TERa is, as predicted, more disordered than C-TERb. Both isoforms have a significant amount of antiparallel β-sheets and a few α-helices, consistent with the folds of the C_2_H_2_ and PHD domains, each consisting of one α-helix and a pair of two antiparallel β-strands. Although the CD spectra revealed distinct fingerprints for the two isoforms, characterising IDPs containing folded domains interspersed with IDRs remains challenging by far-UV CD spectroscopy [85]. Nevertheless, the C-TER regions seem to adopt collapsed structures where β-sheets and α-helices in ZnFs are separated by disordered segments.

Surprisingly, C-TERb and C-TERa display similar aggregation tendencies over the course of time with respect to their distinct structural properties. Indeed, their CD signature gradually shifts to a strong negative band at 227 nm. The rather high value of this band, associated with β-sheet formation, likely arises from twisted β-sheets and a higher proportion of antiparallel β-sheets, though parallel β-sheets are also formed [86]. Such a signature has already been observed for α-syn fibrils [87].

Furthermore, C-TERb and C-TERa display different conformational transformations over time compared to DPF3b and DPF3a FL. Indeed, C-TER aggregation is characterised by an overall increase in parallel at the expense of antiparallel β-sheets, whereas DPF3 FL isoforms only enrich into antiparallel β-sheets. Such enhancement in parallel β-sheets has already been highlighted for other amyloidogenic IDPs. For example, α-syn self-aggregates into parallel β-sheets when heterogeneous nucleation dominates [60,61]. In the first steps of aggregation, the two DPF3 C-TER isoforms rearrange into antiparallel β-sheets by reducing the α-helix content. Interestingly, antiparallel β-sheets are known to be more abundant at the beginning of fibrillation during the formation of high-order oligomers. Noticeably, C-TERb seems to form α-helical intermediates within the first 24 h, which has also been reported for α-syn and, more curiously, for DPF3a FL [17,88]. 

Aggregation-induced structural rearrangement of C-TERs is also evidenced by their intrinsic fluorescence, showing that C-TER fibrillation may be driven by the same aggregation prone region within the 200–261 region. Indeed, according to the absence of the λ_em_ shift in ITF, C-TERb Trp residues do not see their direct environment change. The loss of the Tyr emission shoulder upon C-TERb and C-TERa aggregation, assumed to arise from Tyr residues in the C_2_H_2_ domain, seems to indicate their involvement in the C-TER amyloid core. Tyr residues appear to be brought closer to Asp and/or Glu proton acceptors. Such a decrease in Tyr emission was also detected for DPF3a upon fibrillation. Nevertheless, the potential implication of the carboxy-terminal IDR of C-TERa in its amyloid core cannot be discarded, though it eludes ITyrF analysis due to the absence of Tyr residues in this region. The contribution of Tyr residues in the amyloid core formation is supported by near-UV CD analysis, revealing a similar spectral pattern between C-TER isoforms in the form of a broad positive band from 270 to 300 nm after 96 h. Given that C-TERa has no Trp residues in its sequence, this was assigned to the reorganisation of Tyr residues into a more ordered core upon aggregation. In addition, this band could be amyloid-related, suggesting that the two C-TER isoforms essentially fold into comparable amyloid cores.

Similar to DPF3 FL isoforms, dbAF also occurs in C-TERb and C-TERa aggregates, with the same emission band centred at ~456–460 nm. Interestingly, DPF3b FL and C-TERb exhibit the same subtle fourth dbAF excitation band between 325 and 330 nm, arguing for its specificity to this isoform. We demonstrated that dbAF is an alternative to the ThT binding assay, as it has proven to be a robust and reliable method for describing aggregation kinetics. Indeed, typical three-phase sigmoid-like curves were obtained for each C-TER. The faster C-TERb rearrangement into parallel β-sheets, observed by far-UV CD, can be explained by its shorter growth phase. C-TERa is characterised by a higher dbAF and ThT plateau in its stationary phase, likely corresponding to a larger amount of mature fibrils. Noticeably, C-TERa lag and elongation phases are longer than those of C-TERb, implying that its aggregation pathway mechanistically differs and that longer fibrils may be formed [89]. Although such a plateau discrepancy can also arise from the morphological differences of aggregates, this is not true for the C-TER isoforms. Indeed, TEM has unveiled that they both assemble into very similar flexible 15–20 nm-wide SSFs, which can pack into DSFs and MSFs. While DPF3b and DPF3a were shown to form various structurally aggregates (such as spherical nucleation units acting in the formation of protofibrils, elongating and clustering into striated and twisted ribbon fibrils), much less morphologic diversity has been observed for C-TER isoforms. Moreover, as for DPF3 FL isoforms, the two C-TERs can also aggregate into an amorphous phase and, more sporadically, into TTFs.

As DPF3 is a metalloprotein and the alteration of metal homeostasis is associated with the development of neurodegenerative diseases, we also investigated the influence of divalent metal cations, namely Cu^2+^, Mg^2+^, Ni^2+^, and Zn^2+^, on the aggregation mechanisms and properties of each C-TER. As revealed by far-UV CD, metals do not prevent the rearrangement into β-sheets. While every metal promotes the formation of antiparallel β-sheets (without preventing parallel β-sheets formation) for C-TERb and only Cu^2+^, Ni^2+^, and Zn^2+^ for C-TERa, Mg^2+^ accelerates the enrichment into parallel β-sheets for C-TERa. Because such a tendency participates in C-TER fibrillation, Mg^2+^ is considered as a C-TERa aggregation enhancer. Considering Cu^2+^, Mg^2+^, and Ni^2+^ have been reported to be efficient fluorescence quenchers of free Trp, the C-TERb ITF spectra reveal that they bind to these metals [90,91]. Interestingly, whilst Zn^2+^ cations do not interfere with free Trp emission intensity, they also have a quenching effect, suggesting that other quenching moieties have been brought closer to C-TERb Trp residues upon Zn^2+^ binding.

While Cu^2+^ is the only metal causing a decrease in Tyr band intensity for C-TERb, a similar fluorescence quenching is observed for C-TERa in the presence of Cu^2+^ and Zn^2+^. As Cu^2+^ is known to be a Tyr fluorescence quencher, the C-TER Tyr residues may directly interact with Cu^2+^ cations. Regarding Zn^2+^, it does not hinder the fluorescence of free Tyr, thus indicating that Zn^2+^ binding to C-TERa likely triggers the approach of Tyr quenching groups in its immediate vicinity through tertiary structure reorganisation [32]. The addition of Cu^2+^ could also lead to the same mechanism, as well as Tyr binding to Zn^2+^ is not excluded. Furthermore, as free Tyr fluorescence is not impeded by Mg^2+^ and Ni^2+^, no change in the C-TER Tyr emission intensity shows that Mg^2+^- and Ni^2+^-induced structural modifications are radically different from those of Cu^2+^ and Zn^2+^. Those tendencies appear congruent with the metal-mediated spectral changes observed in far- and near-UV CD. Indeed, the far-UV CD negative β-sheet band is shifted to a lower wavelength for the two C-TER isoforms with Cu^2+^. Specific to C-TERa, Zn^2+^ even further shifts the far-UV CD band. In near-UV CD, the C-TERb and C-TERa Tyr band is also displaced by Cu^2+^ addition, whereas a second absorption band is visible on the C-TERa-Zn^2+^ spectrum. In addition, the C-TERa dbAF signature is affected as well by Cu^2+^ and Zn^2+^ cations, resulting in a new excitation contribution and in the red shift of the emission band, respectively. Taken together, all these results suggest that, compared to Ni^2+^ and Mg^2+^, Cu^2+^ and Zn^2+^ induce specific structural transformations, especially in the neighbourhood of Tyr residues, during C-TERb and C-TERa aggregation, leading to the emergence of unique spectral properties. Nevertheless, additional investigation regarding metal binding, through titration and molecular docking experiments, remains necessary to fully ascertain these hypotheses. 

Significant differences were observed between metal-treated C-TERb and C-TERa in terms of the dbAF kinetics. Each metal inhibits C-TERb fibrillation, leading to an overall dbAF intensity decrease in the following order: Ni^2+^ > Zn^2+^ ≥ Cu^2+^ > Mg^2+^. A decrease in plateau intensity is usually associated with few mature fibrils formed, as well as with different aggregation mechanisms. In the presence of Ni^2+^, the C-TERb elongation phase is significantly extended, advocating for Ni^2+^ promoting and stabilising low-autofluorescent β-sheet oligomers. Although Cu^2+^ and Zn^2+^ have the same kinetics profiles, especially characterised by a longer lag time, their addition may result in structurally different aggregates, as evidenced by CD. Mg^2+^ also reduces the plateau phase, but to a lesser extent. Considering C-TERa, Ni^2+^ and Cu^2+^ have a comparable inhibitory fibrillation effect by extensively increasing the lag and growth phases, as well as reducing the intensity of the stationary phase, promoting oligomer formation. Near-UV CD and ITyrF analyses nonetheless suggest that the inhibition mechanism is different between these two cations. Consistent with the CD spectra, Mg^2+^ significantly enhances the C-TERa fibrillation process, whereas the generation of fibrillation-prone intermediates induced by Zn^2+^ takes more time than metal-free ones.

The dbAF kinetic tendencies were confronted with TEM micrographs, showing that a close link seems to exist between the spectral and morphological properties. Indeed, the C-TERa-Mg^2+^ SSFs retain the width, flexibility, and association ability of those without metal. The spectral uniqueness of Zn^2+^-bound C-TERa is echoed by the remarkable reduction of the fibril width. In the presence of Cu^2+^ and Ni^2+^, C-TERa has a lower quantity of fibrils, which are consistently found rigidified and fragmented into shorter ones. The same peculiarities were observed for C-TERb in each condition tested. Even though broken fibrils were detected for metals, highly populated oligomer fractions have probably not been detected by TEM, because of their smaller size. This is especially true for C-TERb incubated with Ni^2+^, Cu^2+^, and Zn^2+^, as well as for C-TERa with Ni^2+^ and Cu^2+^.

With respect to the TEM data, lower dbAF and ThT emission intensities in the stationary phases most likely also originate from fibril fragmentation and shortening. Indeed, a similar behaviour was notably reported for aggregated tau in the presence of Ni^2+^ [31]. In a similar manner, Co^2+^ and associated complexes are also able to hinder tau aggregation and to destabilise preformed fibrils by breaking them into shorter ones [92]. Consistent with our results, this study highlights the correlation between the fragmentation of tau fibrils and an overall decrease in ThT fluorescence. Such a ThT tendency was also described for laser-induced destruction of keratoepithelin fibrils, with supporting TEM images of short and broken filaments [93]. As fibrils look stiffened in certain conditions, a reduction in ThT emission could also arise from this morphological feature, leading to less ThT binding sites at the fibril surface.

The diversity of metal-induced effects on the aggregation of C-TER isoforms arguably arises from their sequence composition, metal-binding residue distribution, and metal specificity to certain residues. Indeed, while Cu^2+^ and Ni^2+^ have a strong affinity for histidine (His) residues, Mg^2+^ preferentially binds to aspartate (Asp) and glutamate (Glu) residues. Zn^2+^ equally binds to cysteine (Cys) and histidine (His) residues [94,95]. C-TERb is particularly enriched in Cys residues, being the most abundant amino acid in its sequence (Figure 12). While nearly all of them are already involved in Zn^2+^ coordination in C_2_H_2_, PHD-1, and -2 ZnFs, an excess of Zn^2+^ may induce structural rearrangement, resulting in new binding sites. Considering that C-TERa has far fewer Cys residues, the specificity of each C-TER aggregation behaviour to Zn^2+^ cations may arise from this sequence composition difference. The two C-TER isoforms have a comparable number of His residues (most of them located at the same positions), which could explain why Cu^2+^ and Ni^2+^ have close inhibitory fibrillation effects on each isoform. Although the binding sites may be similar, Cu^2+^ and Ni^2+^ have distinct coordination geometries (mostly trigonal plane and octahedral, respectively) [93], which may be associated with the observed spectral differences between these two metals. While C-TERb and C-TERa have equivalent proportions of Asp and Glu residues, they are differentially distributed along their respective sequence towards the C-terminus. Such a discrepancy could notably be responsible for the contrary effects of Mg^2+^ on C-TERb and C-TERa aggregation.

The influence of divalent metal cations (Cu^2+^, Mg^2+^, Ni^2+^, and Zn^2+^) on the aggregation properties of C-TER isoforms has unravelled unique optical–structural property relationships of C-TER aggregates. Gaining insights into the metal-induced aggregation mechanisms of amyloidogenic proteins is essential to better apprehend the pathogenesis of neurodegenerative diseases. Our study shed light on the sensitivity and specificity of the DPF3 C-TER regions to divalent metal cations with respect to their aggregation process. Such knowledge will enhance our understanding of the DPF3 fibrillation pathways in AD and PD. Further work is nonetheless needed to obtain the full picture of DPF3 aggregation. Studying the aggregation propensity of the DPF3 N-terminal region, as well as the effect of monovalent and trivalent metal cations, temperature, or pH could be very informative. Moreover, the susceptibility of DPF3 isoforms to liquid–liquid phase separation should be investigated to better decipher their self-assembly mechanisms.

## 4. Materials and Methods

### 4.1. Overexpression and Purification of C-terminal Regions

Recombinant C-terminal regions (C-TER) of each DPF3 isoform (DPF3b and DPF3a) were expressed with a GST tag at their N-terminus using a pET-like vector in *E. coli* BL21 Rosetta (DE3) cells. Transformed bacterial strains were precultured at 37 °C for 16 to 18 h in 20 g/L of lysogeny Lennox broth (LB) containing 0.36 mM ampicillin. From 5.0 mL of preculture, cultures were grown at 37 °C in 20 g/L LB with 0.14 mM ampicillin until the optical density at 600 nm reached a range of 0.5–0.8. Protein expression was induced at 37 °C for 4 h by the addition of 0.5 mM isopropyl β-D-1-thiogalactopyranoside (IPTG). After induction, cultures were centrifuged, supernatants discarded, and pellets stored at −20 °C. Before purification, pellets were thawed, suspended in lysis buffer (phosphate-buffered saline (PBS) pH 7.3, 0.5% Triton X-100, 200 mM KCl, 200 µM phenylmethylsulfonyl fluoride), and sonicated in an ice-water bath (6 times 30 s with 30 s pauses). Lysates were centrifuged, supernatants were gathered, and proteins were purified using an Äkta Purifier fast protein liquid chromatography (FPLC). GST-fused C-TER proteins were first bound to a 5 mL GSTrap FF prepacked column (Cytiva) in binding buffer (PBS pH 7.3, 200 mM KCl), then cleaved on the column from their tag at 30 °C for 2 h in the presence of 20 µL of TEV protease solution (Sigma) in Tris-buffered saline (TBS; 50 mM Tris-HCl pH 8.0, 150 mM NaCl). After cleavage, the proteins of interest were eluted and recovered in TBS. Their purity and presence in the characterised samples were assessed by sodium dodecyl sulphate polyacrylamide gel electrophoresis (SDS-PAGE) and mass spectrometry. The presented data arose from the analysis of three overexpression and purification batches. As the same tendencies were consistently observed for each batch, only representative results are shown.

### 4.2. Protein Concentration and UV–Visible Absorption Spectroscopy

Purified C-TER domains were further concentrated in TBS using a 6–8 kDa cut-off dialysis membrane rolled up in water-absorbent PEG-20000. UV–visible absorption spectra were recorded with a UV-63000PC spectrophotometer (VWR), using a 10 mm pathlength quartz QS cell (Hellma). Using the B. Kuipers and H. Gruppen method [96], the molar extinction coefficient of C-TER isoforms was calculated (ε_C-TERb_ = 397,435 M^−1^·cm^−1^, ε_C-TERa_ = 257,218 M^−1^·cm^−1^), and their final concentration was determined by measuring the absorbance value at 214 nm. In the conducted experiments, the working protein concentration was ~0.2 mg/mL (~10 µM) for each C-TER region if not explicitly specified. From the C-TER absorption spectra, the foldedness index was calculated using the following equation: foldedness index = (A^280^/A^275^) + (A^280^/A^258^), where A corresponds to the absorbance value at a given wavelength, which is indicated by the corresponding superscript.

### 4.3. Metal Cation and Protein Sample Preparation

Divalent metal cation working solutions (20 mM) were prepared in TBS from anhydrous copper (II) sulphate (CuSO_4_) for Cu^2+^, magnesium sulphate heptahydrate (MgSO_4_∙7 H_2_O) for Mg^2+^, nickel (II) sulphate hexahydrate (NiSO_4_∙6 H_2_O) for Ni^2+^, and zinc sulphate heptahydrate (ZnSO_4_∙7 H_2_O) for Zn^2+^. Protein samples were incubated in the presence of 100 µM of metal cations, by mixing 1 µL of metal cation working solution with 100 µL of purified C-TER domain and 99 µL of TBS (final protein concentration ~0.1 mg/mL or ~5 µM). The protein:metal ratio amounted to approximately 1:20. 

### 4.4. Bioinformatics Analyses

Sequence-based analyses were performed from the known amino acid sequence of DPF3b (Uniprot ID: Q92784-1) and DPF3a (Uniprot ID: Q92784-2) on each of their C-terminal domains: residues 200–378 and 200–357 for C-TERb and C-TERa, respectively. The average predicted per-residue percentage of intrinsic disorder was determined using a set of 18 freely available online disorder predictors: VL-XT, XL1-XT, CaN-XT, VL3, and VSL2 algorithms from Predictors of Natural Disordered Regions (PONDR) [97,98,99]; NMR, X-ray, and Disprot-trained datasets from ESpritz [100]; Protein DisOrder prediction System (PrDOS) [101]; long and short disordered regions from the Prediction of Intrinsically Unstructured Proteins (IUPred3) tool [102]; metapredict (v2.1) deep-learning based predictor [103]; NORSnet, Ucon, and MetaDisorder (MD) algorithms from the PredictProtein webserver [104]; non-evolutionary and evolutionary-based predictors from the Prediction of Order and Disorder by evaluation of NMR data (ODiNPred) [105]. By using the PONDR tool, cumulative distribution function and charge–hydropathy plots were generated [106]. Disordered sequence-associated patterning parameters (κ and Ω) and predicted conformational groups were extracted from Das–Pappu phase diagrams, generated with the Classification of Intrinsically Disordered Ensemble Regions (CIDER) server [45].

### 4.5. Far- and Near-UV Circular Dichroism Spectroscopy 

Far-UV (190–260 nm range) and near-UV (250–320 nm range) CD spectra were recorded with a MOS-500 spectropolarimeter at 20 °C in TBS, using a 1 mm optical pathlength quartz Suprasil cell (Hellma). Four scans were averaged, and buffer baselines were subtracted; corrected spectra were smoothed. In far-UV experiments, the following parameters were used: 30 nm/min scanning rate, 2 nm bandwidth, 0.5 nm data pitch, 1 s digital integration time. In near-UV: 15 nm/min scanning rate, 2 nm bandwidth, 0.5 nm data pitch, 2 s digital integration time. For each spectrum, data are presented as the mean residue ellipticity ([Θ]_MRE_), calculated as follows: [Θ]_MRE_ = (Mθ)/(n − 1)(10γl), where M is the protein molecular mass (Da), θ is the measured ellipticity (mdeg), n is the number of residues in the protein sequence, γ is the protein mass concentration (mg/mL), and l is the cell optical pathlength (cm). Regarding the far-UV CD spectra, the secondary structure content was estimated by using the Beta Structure Selection (BeStSel) online deconvolution webserver in the 200–250 nm range [107].

### 4.6. Intrinsic Fluorescence Spectroscopy (ITF, ITyrF, and dbAF)

Emission spectra of tryptophan (ITF) and tyrosine (ITyrF) residues, as well as deep-blue autofluorescence (dbAF) spectra were recorded from their respective excitation wavelength up to 600 nm with an Agilent Cary Eclipse fluorescence spectrophotometer at ~20 °C in TBS, using a 10 mm optical pathlength quartz GS cell (Hellma). The data pitch was set to 1.0 nm and the excitation and emission slit widths (sw) to 10 nm each. An excitation wavelength (λ_ex_) of 295 nm was used for the ITF experiment, 275 nm for ITyrF, and 400 nm for dbAF. Complementarily to the emission spectra, dbAF excitation spectra were also recorded from 200 nm up to the dbAF emission wavelength with the aforementioned parameters. 

### 4.7. Thioflavin T Binding Assay

The thioflavin T (ThT) working solution (20 µM) was prepared in TBS and filtered on polyether sulfone 0.2 µm. Just before measurement, 75 µL of ThT working solution was added to 75 µL of protein material (final ThT concentration of 10 µM). ThT fluorescence spectra were recorded with an Agilent Cary Eclipse fluorescence spectrophotometer at ~20 °C in TBS, using a 10 mm optical pathlength quartz GS cell (Hellma), a data pitch of 1.0 nm, and excitation and emission sw of 10 nm.

### 4.8. Transmission Electron Microscopy 

Protein aggregates and fibrils were visualised by negative staining on a PHILIPS/FEI Tecnai 10 electron microscope operating at a voltage of 100 kV. Droplets of 5 µL of protein material were left for 3 min on formvar/carbon-coated grids that were previously hydrophilised by glow discharge. Any excess was soaked up with a piece of blotting paper, and the grid was put on a 5 µL droplet of 0.5% (w/v) uranyl acetate for 1 min before being air-dried for 5 min. 

## Figures and Tables

**Figure 1 ijms-23-15291-f001:**
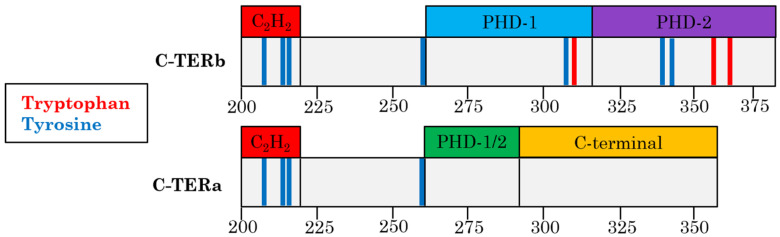
Domains and aromatic residue organisation of C-TERb (**upper** panel) and C-TERa (**lower** panel). C-TERb contains three zinc finger domains (from left to right), namely Krüppel-like C_2_H_2_ (red), PHD-1 (turquoise), and PHD-2 (purple). C-TERa contains (from left to right) one zinc finger, the Krüppel-like C2H2 (red), as well as a truncated PHD finger, PHD-1/2 (green), and a C-terminal domain (yellow). On each schematic sequence, tryptophan residues (Tyr) are represented by red sticks and tyrosine residues (Tyr) by blue ones.

**Figure 2 ijms-23-15291-f002:**
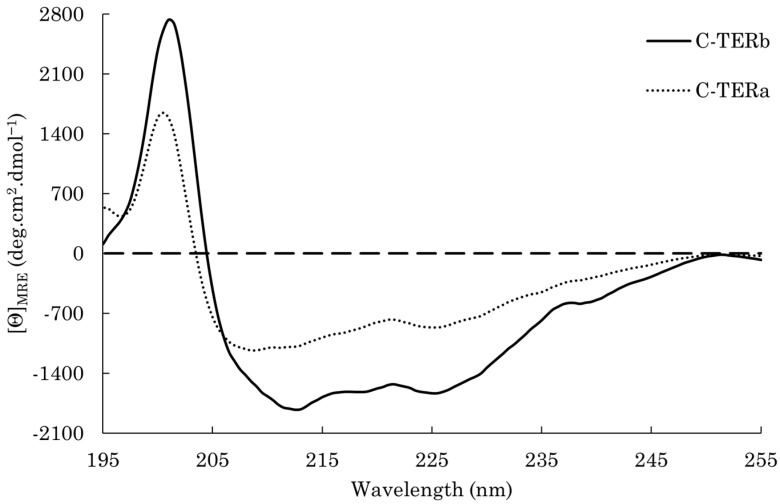
Far-UV CD spectra of C-TERb (solid line) and C-TERa (dotted line) in TBS at ~20 °C (C_C-TER_ = 10 µM).

**Figure 3 ijms-23-15291-f003:**
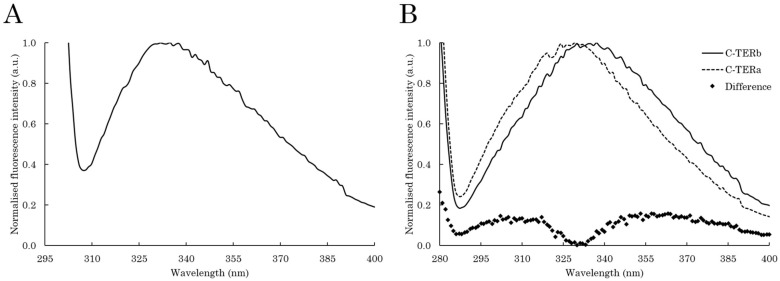
Intrinsic fluorescence spectra of C-TERb and C-TERa (C_C-TER_ = 10 µM) in TBS at ~20 °C. (**A**) Normalised ITF spectrum (λ_ex_ = 295 nm, sw = 10 nm) of C-TERb. (**B**) Normalised ITyrF spectra (λ_ex_ = 275 nm, sw = 10 nm) of C-TERb (solid line) and C-TERa (dashed line) at the associated difference spectrum (diamonds).

**Figure 4 ijms-23-15291-f004:**
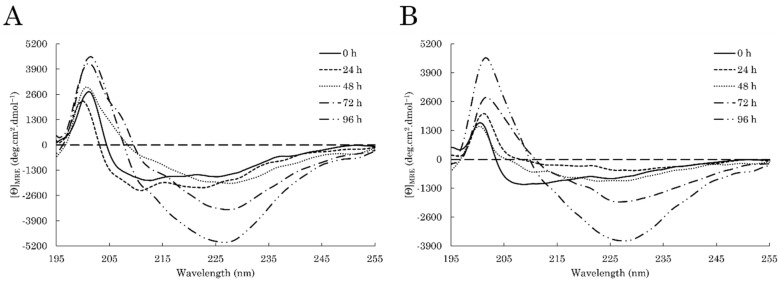
Far-UV CD spectra of (**A**) C-TERb and (**B**) C-TERa (C_C-TER_ = 10 µM) after 0 h (solid line), 24 h (dashed line), 48 h (dotted line), 72 h (dashed-dotted line), and 96 h (dashed-double-dotted line) of incubation in TBS at ~20 °C.

**Figure 5 ijms-23-15291-f005:**
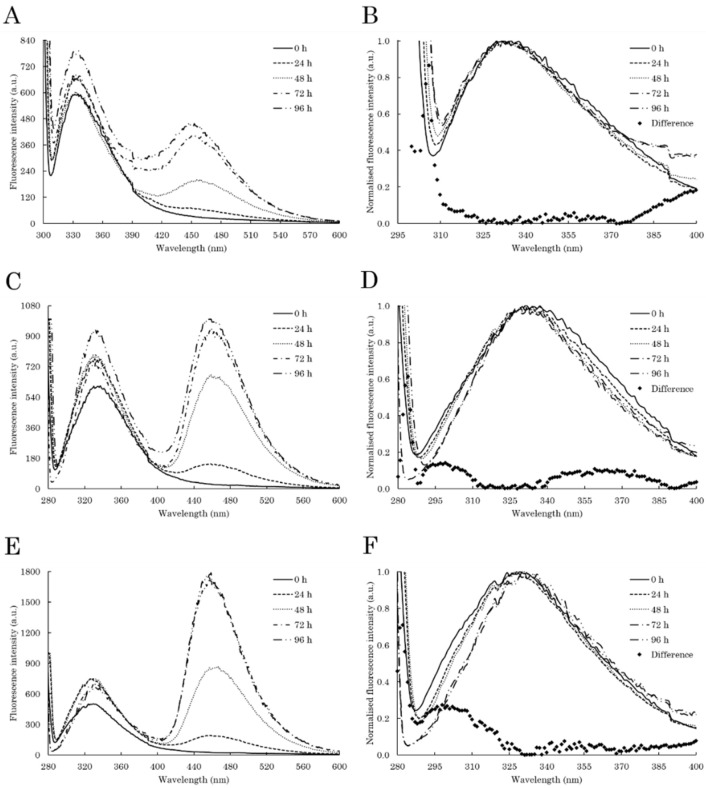
Intrinsic fluorescence spectra of C-TERb (**A**–**D**) and C-TERa (**E**,**F**) after 0 h (solid line), 24 h (dashed line), 48 h (dotted line), 72 h (dashed-dotted line), and 96 h (dashed-double-dotted line) of incubation in TBS at ~20 °C (C_C-TER_ = 10 µM). (**A**) ITF and (**B**) normalised first ITF band spectra (λ_ex_ = 295 nm, sw = 10 nm). (**C**,**E**) ITyrF and (**D**,**F**) normalised first ITyrF band spectra (λ_ex_ = 275 nm, sw = 10 nm). On each normalised spectrum, the associated difference spectrum is determined between 0 and 96 h (diamonds).

**Figure 6 ijms-23-15291-f006:**
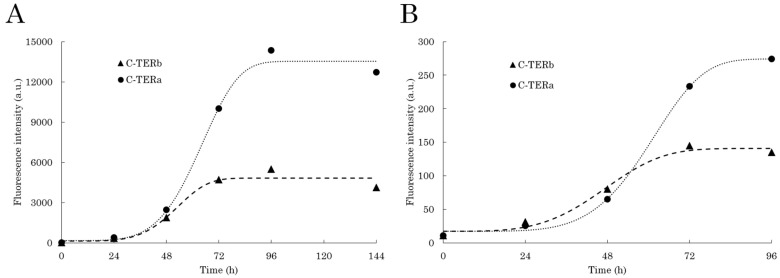
Sigmoid-fitted aggregation kinetic curves of C-TERb (triangles, dashed line) and C-TERa (circles, dotted line), using (**A**) dbAF emission at 456 nm (λ_ex_ = 400 nm, sw = 10 nm) and (**B**) ThT fluorescence at 485 nm (λ_ex_ = 440 nm, sw = 10 nm, C_ThT_ = 10 µM). Samples were incubated in TBS at ~20 °C for 96 to 144 h (C_C-TER_ = 10 µM).

**Figure 7 ijms-23-15291-f007:**
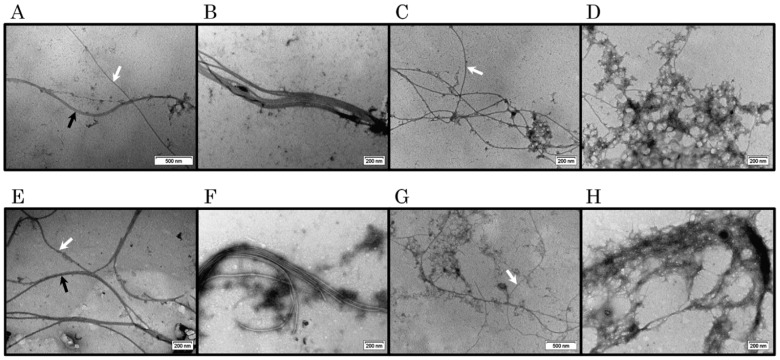
Negatively stained TEM micrographs of (**A**–**D**) C-TERb and (**E**–**H**) C-TERa incubated for 144 h (6 days) in TBS at ~20 °C (C_C-TER_ = 10 µM). The two C-TER isoforms show a similar morphologic diversity. (**A**,**E**) Long single straight (SSFs) (indicated by the white arrows) and double straight fibrils (DSFs) (indicated by the black arrows). (**B**,**F**) Such SSFs and DSFs tend also to densely pack into multiple straight fibrils (MSFs). (**C**,**G**) SSFs (indicated by the white arrows) can be found in extended and diffuse networks of intersecting filaments. (**D**,**H**) Amorphous aggregates are also observed. The scale bar is provided at the bottom right of each TEM micrograph.

**Figure 8 ijms-23-15291-f008:**
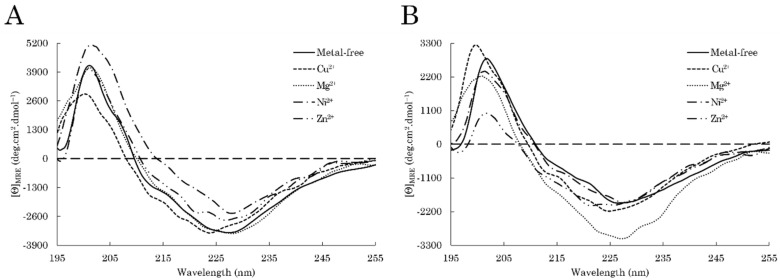
Far-UV CD spectra of (**A**) C-TERb and (**B**) C-TERa (C_C-TER_ = 5 µM) incubated in TBS at ~20 °C for 72 h without metal (solid line) and in the presence of Cu^2+^ (dashed line), Mg^2+^ (dotted line), Ni^2+^ (dashed-dotted line), and Zn^2+^ (dashed-double-dotted line) cations (C_M_^2+^ = 100 µM).

**Figure 9 ijms-23-15291-f009:**
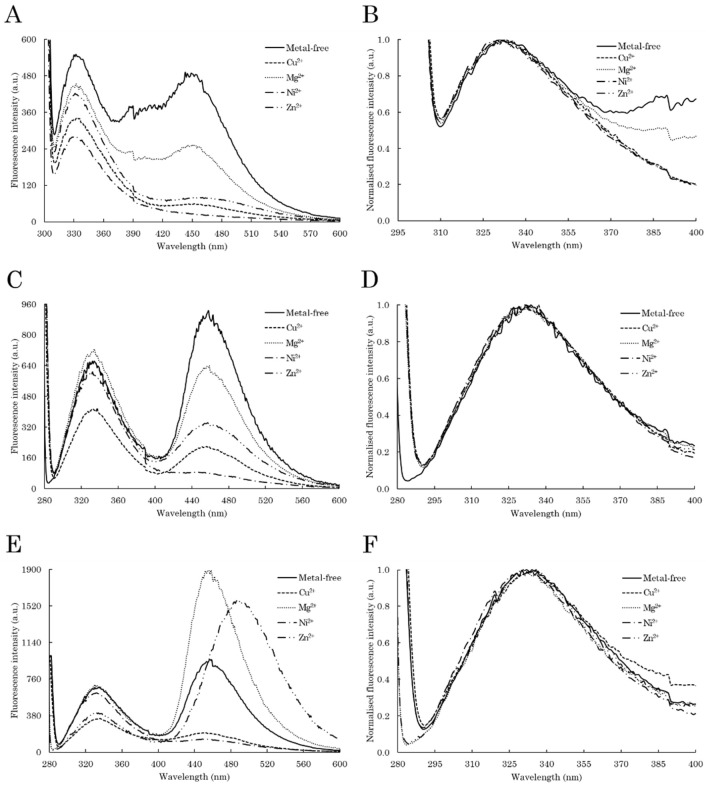
Intrinsic fluorescence spectra of C-TERb (**A**–**D**) and C-TERa (**E**,**F**) incubated in TBS at ~20 °C for 72 h (C_C-TER_ = 5 µM) without metal (solid line) and in the presence of Cu^2+^ (dashed line), Mg^2+^ (dotted line), Ni^2+^ (dashed-dotted line), and Zn^2+^ (dashed-double-dotted line) cations (C_M_^2+^ = 100 µM). (**A**) ITF and (**B**) normalised first ITF band spectra (λ_ex_ = 295 nm, sw = 10 nm). (**C**,**E**) ITyrF and (**D**,**F**) normalised first ITyrF band spectra (λ_ex_ = 275 nm, sw = 10 nm).

**Figure 10 ijms-23-15291-f010:**
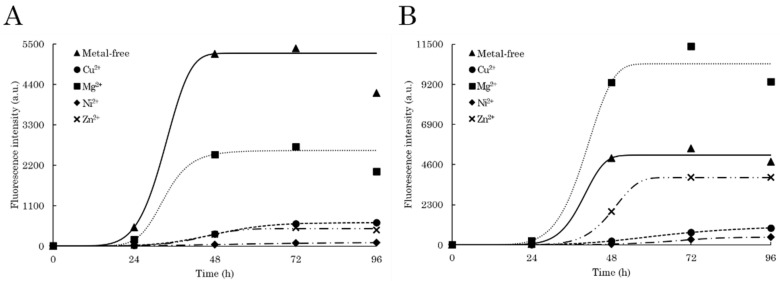
Sigmoid-fitted aggregation kinetic curves of (**A**) C-TERb and (**B**) C-TERa (C_C-TER_ = 5 µM) without metal (triangles, solid line) and in the presence of Cu^2+^ (circles, dashed line), Mg^2+^ (squares, dotted line), Ni^2+^ (diamonds, dashed-dotted line), and Zn^2+^ (crosses, dashed-double-dotted line) cations (C_M_^2+^ = 100 µM), using dbAF emission at 456 nm (λ_ex_ = 400 nm, sw = 10 nm). Samples were incubated in TBS at ~20 °C for 96 h.

**Figure 11 ijms-23-15291-f011:**
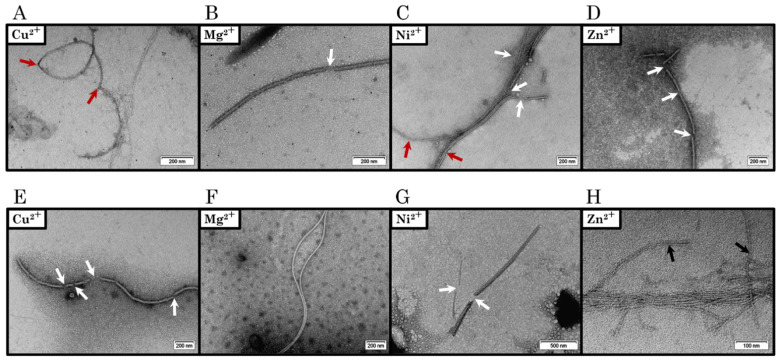
Negatively stained TEM micrographs of (**A**–**D**) C-TERb and (**E**–**H**) C-TERa (C_C-TER_ = 5 µM) incubated for 144 h (6 days) in TBS at ~20 °C in the presence of (**A**,**E**) Cu^2+^, (**B**,**F**) Mg^2+^, (**C**,**G**) Ni^2+^, and (**D**,**H**) Zn^2+^ cations (C_M_^2+^ = 100 µM). Under certain conditions, the formed fibrils present breaking points (indicated by red arrows) and happen to be fragmented (breakages are indicated by white arrows). In Panel H, black arrows pinpoint single straight fibrils (SSFs) that cluster into multiple straight fibrils (MSFs). The scale bar is provided at the bottom right of each TEM micrograph.

**Figure 12 ijms-23-15291-f012:**
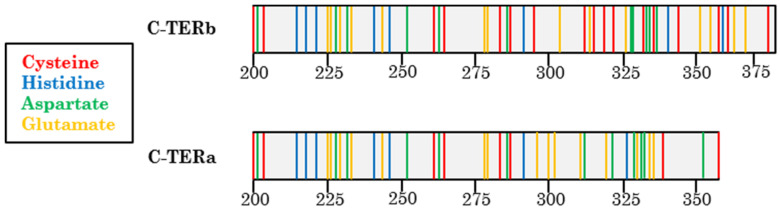
Sequence distribution of amino acids involved in metal binding in C-TERb (**upper** panel) and C-TERa (**lower** panel). Selected residues are the most statistically represented in metal-bound proteins, which include cysteine (Cys), histidine (His), aspartate (Asp), and glutamate (Glu). Cys (red), His (blue), Asp (green), and Glu (yellow) residues are shown as sticks in their associated colour.

**Table 1 ijms-23-15291-t001:** BeStSel secondary structure content estimations and fit-associated RMSD values of C-TERb and C-TERa (C_C-TER_ = 10 µM) incubated in TBS at ~20 °C for 0 h, 24 h, 48 h, 72 h, and 96 h.

		Secondary Structure Content (%)				
C-TER (time)	RMSD	α-Helix	Antiparallel β-Sheet	Parallel β-Sheet	Turn	Coil
C-TERb (0 h)	0.04	4	38	0	17	40
C-TERb (24 h)	0.05	5	35	0	15	46
C-TERb (48 h)	0.05	0	38	3	16	43
C-TERb (72 h)	0.08	0	24	9	21	46
C-TERb (96 h)	0.08	0	24	15	21	40
C-TERa (0 h)	0.03	2	32	0	18	48
C-TERa (24 h)	0.02	0	37	0	18	45
C-TERa (48 h)	0.04	0	37	0	16	47
C-TERa (72 h)	0.06	0	34	5	19	42
C-TERa (96 h)	0.07	0	25	10	22	43

**Table 2 ijms-23-15291-t002:** BeStSel secondary structure content estimations and fit-associated RMSD values of C-TERb and C-TERa (C_C-TER_ = 5 µM) incubated in TBS at ~20 °C for 72 h in the presence of Cu^2+^, Mg^2+^, Ni^2+^, and Zn^2+^ cations (C_M_^2+^ = 100 µM).

		Secondary Structure Content (%)				
C-TER (M^2+^)	RMSD	α-Helix	Antiparallel β-Sheet	Parallel β-Sheet	Turn	Coil
C-TERb	0.08	0	24	9	21	46
C-TERb (Cu^2+^)	0.06	0	31	8	18	43
C-TERb (Mg^2+^)	0.06	0	27	8	20	45
C-TERb (Ni^2+^)	0.07	0	33	6	20	41
C-TERb (Zn^2+^)	0.07	0	30	7	19	44
C-TERa	0.06	0	34	5	19	42
C-TERa (Cu^2+^)	0.05	0	33	5	17	45
C-TERa (Mg^2+^)	0.09	0	21	11	23	45
C-TERa (Ni^2+^)	0.05	0	38	3	18	41
C-TERa (Zn^2+^)	0.06	0	38	3	18	41

## Data Availability

Data will be made available from the corresponding author upon request.

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
