# Peer review of "Unveiling the Metal-Dependent Aggregation Properties of the C-terminal Region of Amyloidogenic Intrinsically Disordered Protein Isoforms DPF3b and DPF3a"

_ijms, 2022, doi:10.3390/ijms232315291_

Round 1

Reviewer 1 Report

In this study, Leyder and colleagues have investigated the disorder, aggregation and fibrillation properties of the C-terminal regions of alpha and beta isoforms (C-TERa, C-TERb) of Double PHD fingers 3 (DPF3). Authors have observed that C-TERa and C-TERb are lacking ordered structures and forming beta-sheet-rich fibrils in a function of time (days). Further, CD and autofluorescence spectroscopy approaches revealed fibrillation-inducer properties of divalent cation metals on C-TERa and C-TERb. Ultimately, authors have employed TEM microscopy to observe the morphological diversification of C-TERa and C-TERb in the absence and presence of divalent cation metals.

The authors are encouraged to consider the following major and minor points to improve the manuscript.

Major points:

1- The authors have particularly tested the effect of divalent cation metals; Have the authors happened to test any monovalent or trivalent metals whether there is any relevance between metal valency and aggregation kinetics, as well as fibril formation?

2- Authors should clearly mention the number of technical and biological replicates: No error bars are plotted for measurements.

3- How is the aggregation to fibrillation transition happening? Some TEM images reveal the co-existence of aggregation and fibrillation; how is the dynamics of this transition regulated?

4- Have the authors happened to check the function of temperature on aggregation and fibrillation kinetics?

5- Do authors know which exact residues are important for metal binding? An anisotropy experiment would be an important input for the manuscript.

6- Have the authors observed whether C-term IDR regions of DPF3a or DPF3b (or together) are phase separating under certain conditions? (e.g. under the function of salt, concentration, temperature, pH)

7- How is the size distribution of fibrils affected in response to w/wo metal addition?

8- Unfortunately, all the experiments are lacking controls. For example, N-term regions of DPF3a and DPF3b are required to clearly credit the C-term IDRs as drivers for aggregation and fibrillation behavior.

Minor point:

9- Scale bars in figures are suggested to be the same for all the images.

Reviewer 2 Report

The manuscript by Leyder et al. analyses the aggregation of C-terminal regions of DPF3b and DPF3a using different spectroscopic methods and TEM imaging. Despite being well-written, there is a lack of clarity and instances of result overinterpretations. I believe multiple points have to be addressed before the manuscript can be considered for publication.

Major points

1.      The authors use BeStSel to determine the distribution of secondary structure elements in the proteins during the aggregation process. This point is purely from personal experience, but the software can have issues when determining the structure of aggregates, as opposed to the native protein state (especially as the data appears to have a significant level of noise – Figure 8). We have had instances where multiple scans and data analysis of the same aggregate sample displayed significantly different secondary structure distributions. In some cases, the software was unable to determine the difference between parallel and antiparallel beta-sheets. This may explain the observed shifts between these two types of beta-sheets.

2.      The signal intensity of dbAF is usually quite small in comparison to Tyr fluorescence, but in the case of Figure 5, it appears to be of equal or even higher intensity. Are the authors sure that this signal originates from dbAF and not some other source, such as bacterial growth in the sample or chemical modifications to one of compounds present in the sample ? Especially since the samples are incubated at room temperature under neutral conditions for multiple days.

3.      The title of the manuscript and the introduction text put an emphasis on the “disordered” parts of the protein, but subsequent CD results and analysis display a significant level of beta-sheet and spiral motifs. Is the nomenclature correct in this case (identifying the proteins in question as IDPs) especially when compared to actual unstructured amyloid proteins, such as Tau or alpha-synuclein ?

4.      Amyloid fibril solutions are known to be very inhomogeneous (samples can contain a large range of different morphology fibrils). The authors display a single TEM image with one or two fibrils for each condition. I do not think this is sufficient enough to make generalized claims about the definitive morphology of the fibrils. As an example, Tau proteins (also an IDP) have been shown to form 4 different morphology fibrils in the same sample.

5.      I would like to criticize the excessive use of acronyms throughout the text. As an example, the first paragraph of the result section contains 25 acronyms. In some cases, the acronym is only used a couple of times (PPID is used 3 times throughout the text and two of the times it is preceded by an explanation of the acronym; OPR is used only twice, with an explanation preceding it). Despite being familiar with the topic of IDPs and amyloids, I spent more time checking the meaning of the acronyms than actually reading the manuscript.

6.      The Results part of the manuscript contains a very large amount of discussion, which is then followed by a Discussion section, which reiterates everything that has been previously explained.

7.      The Result and Discussion sections include far too many assumptions and overstatements, especially when discussing the effect of metal ion binding to proteins, their influence on fluorescence intensity and CD spectra. Such statements would need to be backed up by several additional experiments, including ITC, micro dialysis, molecular docking, titration and so on. Since the manuscript mainly focuses on UV/Vis spectroscopy and TEM imaging, I believe so many assumptions should not be made.

Minor points

1.      There are a couple of instances where there appear to be missing references:

Line 78: “As unraveled in our latest studies”

Lines 90-91 “It is well known...”

1.      Figure 1 contains black text over dark purple and dark blue boxes (PHD-1 and PHD-2). I would advise to choose more contrasting options, as it is difficult to read in a printed, black-white format.

2.      Lines 578-580: this text appears to state that shorter lag times and protein concentration have an unusual correlation. But nucleation events are usually always shorter in the presence of a higher protein concentration, simply due to the occurrence of more interactions between molecules.

Round 2

Reviewer 1 Report

Unfortunately, the authors did not improve the manuscript by experimentally addressing the suggestions to increase the significance of the study. The study certainly requires substantial improvement with a comprehensive revision.

Reviewer 2 Report

The authors have addressed most of the concerns with the manuscript. The addition of TEM images for each condition significantly improves the result and discussion parts of the text. The removal of certain overstatements and the addition of potential future perspectives also nicely contributes to the overall quality of the manuscript.